# Open X-Embodiment: Robotic Learning Datasets and RT-X Models

**Abstract:** Large, high-capacity models trained on diverse datasets have shown remarkable successes on efficiently tackling downstream applications. In domains from NLP to Computer Vision, this has led to a consolidation of pretrained models, with general pretrained backbones serving as a starting point for many applications. Can such a consolidation happen in robotics? Conventionally, robotic learning methods train a separate model for every application, every robot, and even every environment. Can we instead train "generalist" X-robot policy that can be adapted efficiently to new robots, tasks, and environments? In this paper, we provide datasets in standardized data formats and models to make it possible to explore this possibility in the context of robotic manipulation, alongside experimental results that provide an example of effective X-robot policies. We assemble a dataset from 22 different robots collected through a collaboration between 21 institutions, demonstrating 527 skills (160266 tasks). We show that a high-capacity model trained on this data, which we call RT-X, exhibits positive transfer and improves the capabilities of multiple robots by leveraging experience from other platforms.

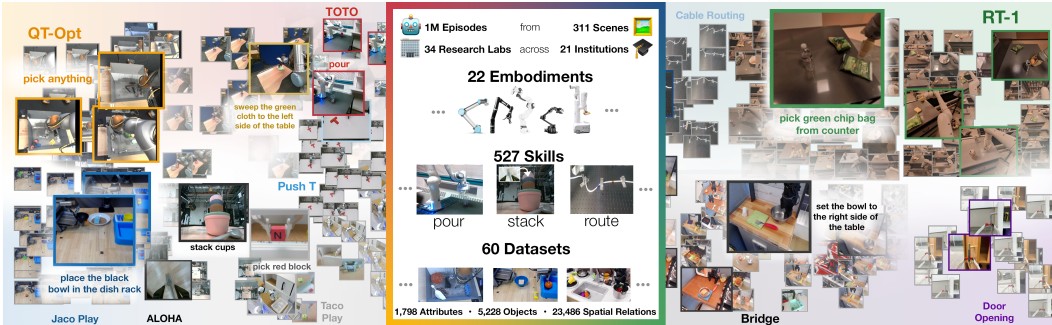

Figure 1: We propose an open, large-scale dataset for robot learning curated from 21 institutions across the globe. The dataset represents diverse behaviors, robot embodiments and environments.

## 1 Introduction

A central lesson from advances in machine learning and artificial intelligence is that large-scale learning from broad and diverse datasets can enable capable AI systems by providing for general-purpose pretrained models. In fact, large-scale general-purpose models typically trained on large and diverse datasets can often outperform their *narrowly targeted* counterparts trained on smaller but more task-specific data. For instance, open-vocabulary image classifiers (e.g., CLIP [1]) trained on large datasets scraped from the web tend to outperform fixed-vocabulary models trained on more limited datasets, and large language models [2, 3] trained on massive text corpora tend to outperform systems that are only trained on narrow task-specific datasets. Increasingly, the most effective way to tackle a given narrow task (e.g., in vision or NLP) is to adapt a general-purpose model. However, these lessons are difficult to apply in robotics: any single robotic domain might be too narrow, and

Submitted to the 7th Conference on Robot Learning (CoRL 2023). Do not distribute.

while computer vision and NLP can leverage large datasets sourced from the web, comparably large and broad datasets for robotic interaction are hard to come by. Even the largest data collection efforts still end up with datasets that are a fraction of the size and diversity of benchmark datasets in vision (5-18M) [4, 5] and NLP (1.5B-4.5B) [6, 7]. More importantly, such datasets are often still narrow along some axes of variation, either focusing on a single environment, a single set of objects, or a narrow range of tasks. How can we overcome these challenges in robotics and move the field of robotic learning toward the kind of large data regime that has been so successful in other domains?

Inspired by the generalization made possible by pretraining large vision or language models on diverse data, we take the perspective that the goal of training generalizable robot policies requires **X-embodiment training**, i.e., with data from multiple robotic platforms. While each individual robotic learning dataset might be too narrow, their union provide a better coverage of variations in environments and robots. Learning generalizable robot policies requires developing methods that can utilize X-embodiment data, tapping into datasets from many labs, robots, and settings. Even if such datasets in their current size and coverage are insufficient to attain the impressive generalization results that have been demonstrated by large language models, in the future, the union of such data can potentially provide this kind of coverage. Because of this, **we believe that enabling research into X-embodiment robotic learning is critical at the present juncture**.

Following this rationale, our work has two goals: **(1)** Demonstrate that policies trained on data from many different robots and environments enjoy the benefits of positive transfer, attaining better performance than policies trained only on data from each evaluation setup. **(2)** Provide datasets, data formats and models for the robotics community to enable future research on X-embodiment models.

Addressing goal **(1)**, we demonstrate that several recent robotic learning methods, with minimal modification, can utilize X-embodiment data and enable positive transfer. Specifically, we train the RT-1 [8] and RT-2 [9] models on 9 different robotic manipulators. We show that the resulting models, which we call RT-X, can improve over policies trained only on data from the evaluation domain, exhibiting better generalization and new capabilities. Addressing **(2)**, we provide the Open X-Embodiment (OXE) Repository, which includes a dataset with 22 different robotic embodiments from 21 different institutions that can enable the robotics community to pursue further research on X-embodiment models, along with open-source tools to facilitate such research. Our aim is not to innovate in terms of the particular architectures and algorithms, but rather to provide the model that we trained together with data and tools to energize research around X-embodiment robotic learning.

## 2  Related Work

**Transfer across embodiments.** A number of prior works have studied methods for transfer across robot embodiments in simulation [10, 11, 12, 13, 14, 15, 16, 17, 18, 19, 20, 21, 22] and on real robots [23, 24, 25, 26, 27, 28, 29]. These methods often introduce mechanisms specifically designed to address the embodiment gap between different robots, such as shared action representations [14, 30], incorporating representation learning objectives [17, 26], adapting the learned policy on embodiment information [30, 31, 11, 18, 15], and decoupling robot and environment representations [24]. Prior work has provided initial demonstrations of X-embodiment training [27] and transfer [25, 32, 29] with transformer models. We investigate complementary architectures and provide complementary analyses, and, in particular, study the interaction between X-embodiment transfer and web-scale pretraining. Similarly, methods for transfer across human and robot embodiments also often employ techniques for reducing the embodiment gap, i.e. by translating between domains or learning transferable representations [33, 34, 35, 36, 37, 38, 39, 40, 41, 42, 43]. Alternatively, some works focus on sub-aspects of the problem such as learning transferable reward functions [44, 17, 45, 46, 47, 48], goals [49], dynamics models [50], or visual representations [51, 52, 53, 54, 55, 56, 57, 58] from human video data. Unlike most of these prior works, we directly train a policy on X-embodiment data, without any mechanisms to reduce the embodiment gap, and observe positive transfer by leveraging that data.

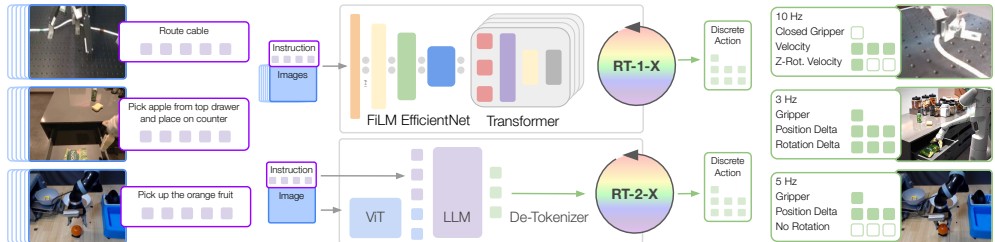

Figure 2: RT-1-X and RT-2-X both take images and a text instruction as input and output discretized end-effector actions. RT-1-X is an architecture designed for robotics, with a FiLM [113] conditioned EfficientNet [114] and a Transformer [115]. RT-2-X builds on a VLM backbone by representing actions as another language, and training action text tokens together with vision-language data.

**Large-scale robot learning datasets.** The robot learning community has created open-source robot learning datasets, spanning grasping [59, 60, 61, 62, 63, 64, 65, 66, 67, 68, 69, 70], pushing interactions [71, 72, 73, 23], sets of objects and models [74, 75, 76, 77, 78, 79, 80, 81, 82, 83, 84], and teleoperated demonstrations [85, 86, 87, 8, 88, 89, 90, 91]. With the exception of RoboNet [23], these datasets contain data of robots of the same type, whereas we focus on data spanning multiple embodiments. The goal of our data repository is complementary to these efforts: we process and aggregate a large number of prior datasets into a single, standardized repository, called Open X-Embodiment, which shows how robot learning datasets can be shared in a meaningul and useful way.

**Language-conditioned robot learning.** Prior work has aimed to endow robots and other agents with the ability to understand and follow language instructions [92, 93, 94, 95, 96, 97], often by learning language-conditioned policies [45, 98, 99, 100, 101, 40, 102, 8]. We train language-conditioned policies via imitation learning like many of these prior works but do so using large-scale multi-embodiment demonstration data. Following previous works that leverage pre-trained language embeddings [103, 45, 104, 99, 40, 105, 106, 8, 107, 108] and pre-trained vision-language models [109, 110, 111, 9] in robotic imitation learning, we study both forms of pre-training in our experiments, specifically following the recipes of RT-1 [8] and RT-2 [9].

# 3   The Open X-Embodiment Repository

We introduce the Open X-Embodiment Repository – an open-source repository which includes **large-scale data** along with **pre-trained model checkpoints** for X-embodied robot learning research. More specifically, we provide and maintain the following open-source resources to the broader community: (1) **Open X-Embodiment Dataset**: robot learning dataset with *1M+ robot trajectories* from 22 *robot embodiments* (2) **Pre-Trained Checkpoints**: a selection of RT-X model checkpoints ready for inference and finetuning.

We intend for these resources to form a foundation for X-embodiment research in robot learning, but they are just the start. Open X-Embodiment is a community-driven effort, currently involving 21 institutions from around the world, and we hope to further broaden participation and grow the initial Open X-Embodiment Dataset over time. The Open X-Embodiment Dataset contains 1M+ real robot trajectories spanning 22 robot embodiments, from single robot arms to bi-manual robots and quadrupeds. The dataset was constructed by pooling 60 *existing* robot datasets from 34 robotic research labs around the world and converting them into a consistent data format for easy download and usage. We use the RLDS data format [112], which saves data in serialized `tfrecord` files and accommodates the various action spaces and input modalities of different robot setups.

# 4   RT-X Design

To evaluate how much X-embodiment training can improve the performance of learned policies on individual robots, we require models that have sufficient capacity to productively make use of such large and heterogeneous datasets. To that end, our experiments will build on two recently proposed

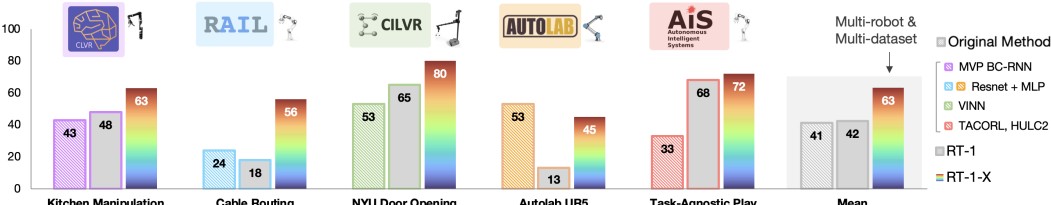

Figure 3: RT-1-X mean success rate is $50\%$ higher than that of either the Original Method or RT-1. RT-1 and RT-1-X have the same network architecture. Therefore the performance increase can be attributed to co-training on the robotics data mixture. The lab logos indicate the physical location of real robot evaluation, and the robot pictures indicate the embodiment used for the evaluation.

Transformer-based robotic policies: RT-1 [8] and RT-2 [9]. We briefly summarize the design of these models in this section, and discuss how we adapted them to the X-embodiment setting in our experiments.

## 4.1 Data format consolidation

One challenge of creating X-embodiment models is that observation and action spaces vary significantly across robots. We use a coarsely aligned action and observation space across datasets. The model receives a history of recent images and language instructions as observations and predicts a 7-dimensional action vector controlling the end-effector ($x$, $y$, $z$, roll, pitch, yaw, and gripper opening or the rates of these quantities). We select one canonical camera view from each dataset as the input image, resize it to a common resolution and convert the original action set into a 7 DoF end-effector action. We normalize each dataset's actions prior to discretization. This way, an output of the model can be interpreted (de-normalized) differently depending on the embodiment used. It should be noted that despite this coarse alignment, the camera observations still vary substantially across datasets, e.g. due to differing camera poses relative to the robot or differing camera properties, see Figure 2. Similarly, for the action space, we do not align the coordinate frames across datasets in which the end-effector is controlled, and allow action values to represent either absolute or relative positions or velocities, as per the original control scheme chosen for each robot. Thus, the same action vector may induce very different motions for different robots.

## 4.2 Policy architectures

We consider two model architectures in our experiments: (1) RT-1 [8], an efficient Transformer-based architecture designed for robotic control, and (2) RT-2 [9] a large vision-language model co-fine-tuned to output robot actions as natural language tokens. Both models take in a visual input and natural language instruction describing the task, and output a tokenized action. For each model, the action is tokenized into 256 bins uniformly distributed along each of eight dimensions; one dimension for terminating the episode and seven dimensions for end-effector movement. Although both architectures are described in detail in their original papers [8, 9], we provide a short summary of each below:

**RT-1 [8]** is a 35M parameter network built on a Transformer architecture [115] and designed for robotic control, as shown in Fig. 2. It takes in a history of 15 images along with the natural language. Each image is processed through an ImageNet-pretrained EfficientNet [114] and the natural language instruction is transformed into a USE [116] embedding. The visual and language representations are then interwoven via FiLM [113] layers, producing 81 vision-language tokens. These tokens are fed into a decoder-only Transformer, which outputs the tokenized actions.

**RT-2 [9]** is a family of large vision-language-*action* models (VLAs) trained on Internet-scale vision and language data along with robotic control data. RT-2 casts the tokenized actions to text tokens, e.g., a possible action may be "1 128 91 241 5 101 127". As such, any pretrained vision-language model (VLM [117, 118, 119]) can be finetuned for robotic control, thus leveraging the backbone of VLMs and transferring some of their generalization properties. In this work, we focus on the

| Evaluation Setting | Bridge | Bridge | RT-1 paper 6 skills |
|---|---|---|---|
| Evaluation Location | IRIS (Stanford) | RAIL Lab (UCB) | Google Robotic Lab |
| Robot Embodiment | WidowX | WidowX | Google Robot |
| Original Method | LCBC [122] | LCBC [122] | - |
| Original Method | 13% | 13% | - |
| RT-1 | 40% | **30%** | **92%** |
| RT-1-X | 27% | 27% | 73% |
| RT-2-X (55B) | **50%** | **30%** | **91%** |

Table 1: Parameter count scaling experiment to assess the impact of capacity on absorbing large-scale diverse embodiment data. For these large-scale datasets (Bridge and RT-1 paper data), RT-1-X underfits and performs worse than the Original Method and RT-1. RT-2-X model with significantly many more parameters can obtain strong performance in these two evaluation scenarios.

RT-2-PaLI-X variant [117] built on a backbone of a visual model, ViT [120], and a language model, UL2 [121], and pretrained primarily on the WebLI [117] dataset.

### 4.3 Training and inference details

Both models use a standard categorical cross-entropy objective over their output space (discrete buckets for RT-1 and all possible language tokens for RT-2).

We define the robotics data mixture used across all of the experiments as the data from 9 manipulators, and taken from RT-1 [8], QT-Opt [65], Bridge [122], Task Agnostic Robot Play [123, 124], Jaco Play [125], Cable Routing [126], RoboTurk [127], NYU VINN [128], Austin VIOLA [129], Berkeley Autolab UR5 [130], TOTO [131] and Language Table [88] datasets. RT-1-X is trained on only robotics mixture data defined above, whereas RT-2-X is trained via co-fine-tuning (similarly to the original RT-2 [9]), with an approximately one to one split of the original VLM data and the robotics data mixture. Note that the robotics data mixture used in our experiments includes 9 embodiments which is fewer than the entire Open X-Embodiment dataset (22) – the practical reason for this difference is that we have continued to extend the dataset over time, and at the time of the experiments, the dataset above represented all of the data. In the future, we plan to continue training policies on the extended versions of the dataset as well as continue to grow the dataset together with the robot learning community.

At inference time, each model is run at the rate required for the robot (3-10 Hz), with RT-1 run locally and RT-2 hosted on a cloud service and queried over the network.

## 5 Experimental Results

Our experiments answer three questions about the effect of X-embodiment training: (1) Can policies trained on our X-embodiment dataset effectively enable positive transfer, such that co-training on data collected on multiple robots improves performance on the training task? (2) Does co-training models on data from multiple platforms and tasks improve generalization to new, unseen tasks? (3) What is the influence of different design dimensions, such as model size, model architecture or dataset composition, on performance and generalization capabilities of the resulting policy? To answer these questions we conduct the total number of 3600 evaluation trials across 6 different robots.

### 5.1 In-distribution performance across different embodiments

To assess the ability of our RT-X model variants to learn from X-embodiment data, we evaluate their performance on in-distribution tasks. We split our evaluation into two types of use cases: evaluation on domains that only have small-scale datasets (Fig. 3), where we would expect transfer from larger datasets to significantly improve performance, and evaluation on domains that have large-scale datasets (Table 1), where we expect further improvement to be more challenging. Note that we use the same robotics data *training* mixture (defined in Sec. 4.3) for all the evaluations presented in this section. For small-scale dataset experiments, we consider Kitchen Manipulation [125], Cable Routing [126], NYU Door Opening [128], AUTOLab UR5 [130], and Robot Play [132]. We use

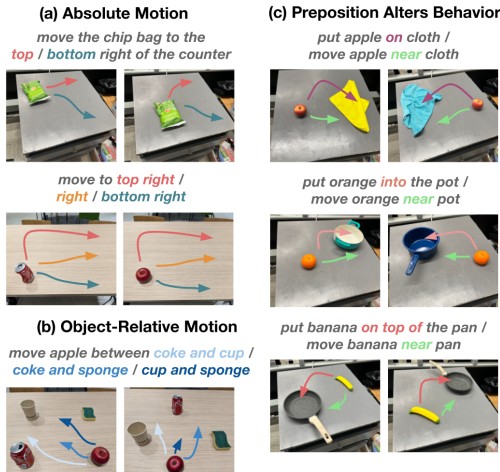

Figure 4: To assess transfer *between* embodiments, we evaluate the RT-2-X model on out-of-distribution skills. These skills are in the Bridge dataset, but not in the Google Robot dataset (the embodiment they are evaluated on).

the same evaluation and robot embodiment as in the respective publications. For large-scale dataset experiments, we consider Bridge [122] and RT-1 [8] for in-distribution evaluation and use their respective robots: WidowX and Google Robot.

For each small dataset domain, we compare the performance of the RT-1-X model, and for each large dataset we consider both the RT-1-X and RT-2-X models. For all experiments, the models are co-trained on the full X-embodiment dataset. Throughout this evaluation we compare with two baseline models: (1) The model developed by the creators of the dataset trained only on that respective dataset. This constitutes a reasonable baseline insofar as it can be expected that the model has been optimized to work well with the associated data; we refer to this baseline model as the *Original Method* model. (2) An RT-1 model trained on the dataset in isolation; this baseline allows us to assess whether the RT-X model architectures have enough capacity to represent policies for multiple different robot platforms simultaneously, and whether co-training on multi-embodiment data leads to higher performance.

**Small-scale dataset domains** (Fig. 3). RT-1-X outperforms Original Method trained on each of the robot-specific datasets on 4 of the 5 datasets, with a large average improvement, demonstrating domains with limited data benefit substantially from co-training on X-embodiment data.

**Large-scale dataset domains** (Table 1). In the large-dataset setting, the RT-1-X model does not outperform the RT-1 baseline trained on only the embodiment-specific dataset, which indicates underfitting for that model class. However, the larger RT-2-X model outperforms both the Original Method and RT-1 suggesting that X-robot training can improve performance in the data-rich domains, but only when utilizing a sufficiently high-capacity architecture.

## 5.2  Improved generalization to out-of-distribution settings

We now examine how X-embodiment training can enable better generalization to out-of-distribution settings and more complex and novel instructions. These experiments focus on the high-data domains, and use the RT-2-X model.

**Unseen objects, backgrounds and environments.** We first conduct the same evaluation of generalization properties as proposed in [9], testing for the ability to manipulate unseen objects in unseen environments and against unseen backgrounds. We find that RT-2 and RT-2-X perform roughly on par (Table 2, rows (1) and (2), last column). This is not unexpected, since RT-2 already generalizes well (see [9]) along these dimensions due to its VLM backbone.

**Emergent skills evaluation.** To investigate the transfer of knowledge across robots, we conduct experiments with the Google Robot, assessing the performance on tasks like the ones shown in Fig. 4.

| Row | Model | Size | History Length | Dataset | Co-Trained Web | Initial Checkpoint | Emergent Skills Evaluation | RT-2 Generalization Evaluation |
|-----|-------|------|---------------|---------|---------------|--------------------|--------------------------|-------------------------------|
| (1) | RT-2 | 55B | none | Google Robot action | Yes | Web-pretrained | 27.3% | **62%** |
| (2) | RT-2-X | 55B | none | Robotics data | Yes | Web-pretrained | **75.8%** | 61% |
| (3) | RT-2-X | 55B | none | Robotics data except Bridge | Yes | Web-pretrained | 42.8% | 54% |
| (4) | RT-2-X | 5B | 2 | Robotics data | Yes | Web-pretrained | 44.4% | 52% |
| (5) | RT-2-X | 5B | none | Robotics data | Yes | Web-pretrained | 14.5% | 30% |
| (6) | RT-2-X | 5B | 2 | Robotics data | No | From scratch | 0% | 1% |
| (7) | RT-2-X | 5B | 2 | Robotics data | No | Web-pretrained | 48.7% | 47% |

Table 2: Ablations to show the impact of design decisions on generalization (to unseen objects, backgrounds, and environments) and emergent skills (skills from other datasets on the Google Robot), showing the importance of Web-pretraining, model size, and history.

These tasks involve objects and skills that are not present in the RT-2 dataset but occur in the Bridge dataset [122] for a different robot (the *WidowX robot*). Results are shown in Table 2, Emergent Skills Evaluation column. Comparing rows (1) and (2), we find that RT-2-X outperforms RT-2 by $\sim 3\times$, suggesting that incorporating data from other robots into the training improves the range of tasks that can be performed even by a robot that already has large amounts of data available. Our results suggest that co-training with data from other platforms imbues the RT-2-X controller with additional skills for the platform that are not present in that platform's original dataset.

Our next ablation involves removing the Bridge dataset from RT-2-X training: Row (3) shows the results for RT-2-X that includes all data used for RT-2-X except the Bridge dataset. This variation significantly reduces performance on the hold-out tasks, suggesting that transfer from the *WidowX* data may indeed be responsible for the additional skills that can be performed by RT-2-X with the Google Robot.

## 5.3 Design decisions

Lastly, we perform ablations to measure the influence of different design decisions on the generalization capabilities of our most performant RT-2-X model, which are presented in Table 2. We note that including a short history of images significantly improves generalization performance (row (4) vs row (5)). Similarly to the conclusions in the RT-2 paper [9], Web-based pre-training of the model is critical to achieving a high performance for the large models (row (4) vs row (6)). We also note that the $55B$ model has significantly higher success rate in the Emergent Skills compared to the $5B$ model (row (2) vs row (4)), demonstrating that higher model capacity enables higher degree of transfer across robotic datasets. Contrary to previous RT-2 findings, co-fine-tuning and fine-tuning have similar performance in both the Emergent Skills and Generalization Evaluation (row (4) vs row (7)), which we attribute to the fact that the robotics data used in RT-2-X is much more diverse than the previously used robotics datasets.

## 6 Discussion, Future Work, and Open Problems

We presented a consolidated dataset that combines data from 22 robotic embodiments collected through a collaboration between 21 institutions, demonstrating 527 skills (160266 tasks). We also presented an experimental demonstration that Transformer-based policies trained on this data can exhibit significant positive transfer between the different robots in the dataset. Our results showed that the RT-1-X policy has a $50\%$ higher success rate than the original, state-of-the-art methods contributed by different collaborating institutions, while the bigger vision-language-model-based version (RT-2-X) demonstrated $\sim 3\times$ generalization improvements over a model trained only on data from the evaluation embodiment. In addition, we provided multiple resources for the robotics community to explore the X-embodiment robot learning research, including: the unified X-robot and X-institution dataset, sample code showing how to use the data, and the RT-1-X model to serve as a foundation for future exploration.

While RT-X demonstrates a step towards a X-embodied robot generalist, there are many more steps needed to make this future a reality. Our experiments have a number of limitations: it does not

consider robots with very different sensing and actuation modalities, it does not study generalization to new robots, and it does not provide a decision criterion for when positive transfer does or does not happen. Studying these questions is an important direction for future work. We hope that this work will serve not only as an example that X-robot learning is feasible and practical, but also provide the tools to advance research in this direction in the future.

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
