# OpenReview forum: "Open X-Embodiment: Robotic Learning Datasets and RT-X Models"
_robot-learning.org/CoRL/2023/Workshop/TGR — CoRL 2023 Workshop TGR Oral_

### Official Review · Reviewer_E2TM · 2023-10-16

**Rating:** 9
**Confidence:** 3

**Review:**

This paper provides a dataset with various skills/tasks for training generalist robots. The carefully collected dataset is very relevant to this workshop's topic and will potentially make important contribution to the entire community.

---

### Official Review · Reviewer_5ZM7 · 2023-10-20

**Rating:** 10
**Confidence:** 4

**Review:**

This paper has compiled an extensive dataset of robot manipulation data encompassing a wide range of skills, potentially being one of the most substantial datasets to date. It offers significant contributions that align well with the theme of this workshop.

---

### Decision · Program_Chairs · 2023-10-20

**Decision:**

Accept (Oral)

**Comment:**

Amazing effort collecting and combining such a large scale robotic dataset. Could be a crucial step towards generalist robot learning.